# Impact of Pesticide Type and Emulsion Fat Content on the Bioaccessibility of Pesticides in Natural Products

**DOI:** 10.3390/molecules25061466

**Published:** 2020-03-24

**Authors:** Ruojie Zhang, Zipei Zhang, Ruyi Li, Yunbing Tan, Shanshan Lv, David Julian McClements

**Affiliations:** 1Department of Food Science, University of Massachusetts Amherst, Amherst, MA 01003, USA; ruojie@mit.edu (R.Z.); zipei.zhang@childrens.harvard.edu (Z.Z.); mwlry19901016@163.com (R.L.); ytan@umass.edu (Y.T.); shanshan2019@nwafu.edu.cn (S.L.); 2Key Laboratory of Food Science and Technology, School of Food, Nanchang University, Nanchang 330047, Jiangxi, China; 3College of Forestry, Northeast A&F University, Yangling 712100, Shaanxi, China

**Keywords:** nanoemulsions, pesticides, Log P values, lipid content, bioaccessibility

## Abstract

There is interest in incorporating nanoemulsions into certain foods and beverages, including dips, dressings, drinks, spreads, and sauces, due to their potentially beneficial attributes. In particular, excipient nanoemulsions can enhance the bioavailability of nutraceuticals in fruit- and vegetable-containing products consumed with them. There is, however, potential for them to also raise the bioavailability of undesirable substances found in these products, such as pesticides. In this research, we studied the impact of excipient nanoemulsions on the bioaccessibility of pesticide-treated tomatoes. We hypothesized that the propensity for nanoemulsions to raise pesticide bioaccessibility would depend on the polarity of the pesticide molecules. Bendiocarb, parathion, and chlorpyrifos were therefore selected because they have Log P values of 1.7, 3.8, and 5.3, respectively. Nanoemulsions with different oil contents (0%, 4%, and 8%) were fabricated to study their impact on pesticide uptake. In the absence of oil, the bioaccessibility increased with increasing pesticide polarity (decreasing Log P): bendiocarb (92.9%) > parathion (16.4%) > chlorpyrifos (2.8%). Bendiocarb bioaccessibility did not depend on the oil content of the nanoemulsions, which was attributed to its relatively high water-solubility. Conversely, the bioaccessibility of the more hydrophobic pesticides (parathion and chlorpyrifos) increased with increasing oil content. For instance, for chlorpyrifos, the bioaccessibility was 2.8%, 47.0%, and 70.7% at 0%, 4%, and 8% oil content, respectively. Our findings have repercussions for the utilization of nanoemulsions as excipient foods in products that may have high levels of undesirable non-polar substances, such as pesticides.

## 1. Introduction

Emulsion-based foods are commonly consumed with fruits and vegetables, e.g., dressings and dips with salads, hot sauces with cooked vegetables, ice creams with fruit pies, whipped creams with fruit salads, and pouring creams with fruit smoothies [1,2]. These types of emulsion-based products are primarily consumed to enhance the palatability of fruits and vegetables, but they may also enhance their nutritional profile by boosting the bioavailability of the lipophilic nutraceuticals contained within them [3,4,5,6,7,8,9]. For instance, the presence of digestible lipids within an emulsion increases the bioaccessibility of lipophilic nutraceuticals by forming mixed micelles capable of solubilizing and carrying them [10,11,12,13,14,15,16]. The bioavailability-boosting effects of emulsions are influenced by the type of ingredients used to assemble them, their structural organization, and the intrinsic properties of the nutraceuticals (e.g., molecular dimensions and hydrophobicity) [17,18,19].

One concern with utilizing emulsions as excipient foods to enhance the bioavailability of nutraceuticals in agricultural products, is that they could also increase the bioavailability of detrimental substances (like pesticides). Pesticides are commonly applied to fruits and vegetables during their growth and/or postharvest processing [20,21,22]. Previously, we demonstrated that mixing emulsions with tomatoes prior to digestion increased the bioaccessibility of chlorpyrifos, a non-polar pesticide [23]. As mentioned earlier, the digestion of oil droplets results in the production of mixed micelles within the gastrointestinal fluids, which solubilize any non-polar substances and carry them to the intestinal lining where they can be absorbed [10,11]. Previous studies have shown that the molecular characteristics of beneficial bioactive agents (nutraceuticals) impacts their bioaccessibility [24]. In particular, the molecular dimensions and hydrophobicity of bioactive agents impacts their solubility in the gastrointestinal fluids [25]. Typically, more non-polar bioactives are solubilized within the non-polar core of the mixed micelles, whereas more polar ones are directly solubilized in the surrounding water [24]. Even so, the non-polar bioactives must still have molecular dimensions that allow them to be incorporated into the mixed micelles: If they are too long, then they will not fit into the non-polar core, leading to a low bioaccessibility [25,26]. For the current study, it was hypothesized that pesticide bioaccessibility would also be influenced by their molecular characteristics, especially their hydrophobicity, as represented by their Log P value.

For this reason, the influence of pesticide type on its bioaccessibility in tomatoes co-ingested with a model food emulsion was examined. Three representative pesticides with different Log P values were examined: Bendiocarb (Log P = 1.7); parathion (Log P = 3.8); and chlorpyrifos (Log P = 5.3). The bioaccessibilities of the pesticides were measured after the samples were passed through a simulated gastrointestinal tract (GIT) [23]. Samples containing different amounts of oil droplets (0%, 4%, and 8%) were used to study the impact of the oil content of emulsion-based foods on pesticide bioaccessibility. Our results provide important insights into the influence of the intrinsic properties of pesticides on their bioaccessibility in the presence of emulsion-based foods.

## 2. Results and Discussion

### 2.1. Initial Emulsion Characteristics

Stock emulsions were prepared using a microfluidizer and then their particle characteristics were determined before they were mixed with the tomato puree. The lipid droplets produced were relatively small (*d_32_* ≈ 0.14 μm) and highly anionic (≈ –66 mV). The small droplet dimensions indicate that there was sufficient emulsifier present and the microfluidization procedure was efficient, while the large negative charge was because the measurement pH (pH 7) was considerably higher than the isoelectric point of the adsorbed whey proteins (pH 5) [27,28].

### 2.2. Impact of GIT Conditions on Particle Properties

Emulsions with different lipid levels (4% or 8%) or buffer solutions without lipids (0%) were mixed with the tomato puree. The properties of the particles in these mixtures were then measured after they were sequentially exposed to simulated GIT conditions: mouth, stomach, and small intestine. The application of a small amount of pesticide to the samples did not affect their particle properties (size, charge, and microstructure) under GIT conditions. For this reason, we only show the results for the samples treated with bendiocarb in this section.

#### 2.2.1. Initial Samples

The mean particle diameter of all the initial samples was relatively high (>20 μm) (Figure 1), which can be attributed to the presence of the large plant-tissue fragments from the tomato puree. The particle size distribution measurements indicated that the samples containing only tomato had a monomodal distribution, while the ones containing both lipid droplets and tomato had a bimodal distribution (Figure 2). The dimensions of the large particles in the mixed systems is consistent with tomato fragments, while the dimensions of the small particles is consistent with oil droplets. The confocal microscopy images suggest that the oil droplets had become highly flocculated after they were mixed with the tomato tissues (Figure 3), which may have been due to a change in solution conditions. For instance, there may have been biopolymers or mineral ions released from the tomato tissues that promoted aggregation of the protein-coated droplets [29,30].

In the absence of emulsion (0% oil), the initial tomato samples had a fairly high negative charge (–31 mV) (Figure 4), which may have been due to anionic colloidal particles or biopolymers, such as plant-cell fragments or pectin. The magnitude of the negative charge was considerably lower in the mixed systems (–21.1 and −14.5 mV) than in the emulsions alone (–66 mV). This phenomenon may be the result of charged substances associated with the tomato tissues, such as plant cell fragments, organic acids, mineral ions, or biopolymers [29,30]. These substances may have contributed to the electrophoresis signal used to determine the ζ-potential of the mixed samples. Alternatively, they may have absorbed to the surfaces of the lipid droplets or increased the ionic strength of the surrounding solution [25], thereby reducing the magnitude of the ζ-potential.

#### 2.2.2. Mouth

In the absence of emulsion (0% oil), the measured particle size of the tomato tissue remained relatively high after exposure to oral conditions (Figure 1). Conversely, in the presence of emulsion (4% and 8% oil), the particle size increased appreciably. The particle size distribution measurements suggest that the peak associated with the oil particles disappeared after mixing with simulated saliva (Figure 2). Interestingly, confocal microscopy analysis suggested that the flocs formed in the initial samples became disrupted after exposure to oral conditions (Figure 3). This suggests that the mucin or other components in the simulated saliva were able to breakdown the flocs. Presumably, the individual oil droplets scattered light much less efficiently than the large flocs or tomato fragments, and so they were not observed in the particle size measurements. Consequently, the measured particle size corresponds to that of the large tomato fragments, rather than the overall system.

In the absence of emulsion, the magnitude of the negative charge on the tomato samples decreased (–25 mV) somewhat compared to the initial sample (–31 mV) (Figure 4). This result may have been because mineral ions in the saliva screened the electrostatic interactions. Alternatively, the anionic mucin molecules may have contributed to the electrophoretic signal used to calculate the ζ-potential. In the presence of emulsion (4% and 8% oil), the negative surface potential of the mixed samples after exposure to the mouth phase (−45.9 and −44.9 mV) was considerably higher than that of the initial samples (−21.1 and −14.5 mV). However, the negative charge was still less than that reported for the oil droplets without tomato fragments (–66 mV). These findings are consistent with the individual oil droplets making a bigger contribution to the electrophoresis signal used in the mouth, presumably because the flocs had broken down. The fact that the surface potential was less than that of the individual oil droplets in the original emulsions may have been due to mineral ions or mucin within the artificial saliva [31,32].

#### 2.2.3. Stomach

In the absence of oil droplets, the mean particle diameter remained high suggesting that the plant-tissue in the tomato puree was resistant to highly acidic gastric conditions (Figure 2). In the mixed samples containing oil droplets, a pronounced reduction in the mean particle diameter occurred (Figure 2), which was due to droplet flocculation within the gastric fluids (Figure 3; Figure 4). This result appears contradictory, since droplet flocculation would be expected to increase the particle size. This is, however, probably an artefact of the light scattering technique. Flocculated droplets scatter light more efficiently than non-flocculated ones, and so they make a bigger contribution to the overall light scattering pattern used to determine the PSD. The flocculation of the protein-coated oil droplets under gastric conditions seen here agrees with the results of our previous study [33]. This type of gastric-flocculation has been linked to various physiochemical processes that may act in concert: (1) The protein layers coating the oil droplets were partially digested by pepsin, allowing the droplets to come closer together; (2) the protein-coated oil droplets became positively-charged under acidic gastric conditions, inducing bridging by negatively-charged mucin molecules; (3) the magnitude of surface potential on the protein-coated droplets decreased because salt ions in the gastric fluids weakened the electrostatic repulsion between them; and (4) non-adsorbed mucin molecules promoted depletion flocculation by generating an osmotic stress [34,35].

The surface potential of all the samples changed from negative in the mouth to slightly positive in the stomach (Figure 4). In the absence of emulsion, this effect can be attributed to the protonation of the carboxyl groups on the tomato fragments at low pH values. In the presence of emulsion, this phenomenon is mainly because the protein-coated oil droplets become positively-charged under gastric conditions [34,35]. As a result, anionic mucin molecules adsorb to their surfaces, leading to charge neutralization. 

#### 2.2.4. Small Intestine

In the absence of emulsion, the mean particle diameter remained relatively high, suggesting that the plant cell fragments in the tomato puree were also resistant to small intestine conditions. In the presence of emulsion, there was a slight increase in mean particle diameter compared to the stomach samples (Figure 1). Conversely, the particle size distribution (Figure 2) and confocal microscopy images (Figure 3) indicated that the majority of the lipid droplets in the emulsions had been digested by lipase. The observed increase in particle size can therefore be attributed to the fact that only the large plant-cell tissue fragments from the tomato puree remained in the samples at the end of the small intestine phase. These results show that the lipase is still able to efficiently digest the oil droplets in the presence of tomato tissue. We must stress that the small intestinal fluids contain numerous kinds of colloidal particles, including micelles, liposomes, insoluble calcium soaps, undigested oil droplets, and plant cell fragments, making it difficult to accurately interpret the light scattering results. 

The ζ-potential of all the samples was strongly negative at the end of the small intestine phase (Figure 4). This effect is mainly because the different kinds of colloidal particles in the small intestinal fluids are assembled from anionic molecular species, such as bile acids, free fatty acids, and peptides [29].

### 2.3. Lipid Digestion Profiles

The influence of oil droplet concentration on lipid digestion was monitored using the pH-stat method. The volume of NaOH solution needed to neutralize the free fatty acids generated by lipid digestion was monitored (Figure 5a) and then used to estimate the percentage of free fatty acids released (Figure 5b). For the control sample (no oil droplets), the volume of NaOH added to the samples during the small intestine period only increased slightly, which may have been due to hydrolysis of proteins, phospholipids, or other digestible substances in the tomato puree. In the presence of oil droplets, there was a steep rise in the volume of NaOH added during the first 20 min, followed by a gradual rise at longer times. As expected, the total amount of NaOH added to the samples was greater for the samples containing 8% oil than the ones containing 4% oil, because there were more triacylglycerol molecules available for hydrolysis.

The percentage of free fatty acids released from the samples containing oil droplets was estimated from the NaOH volume added, after subtracting the value for the control (0% oil). After 2 h, 104.0% and 77.1% of the oil phase in the samples initially containing 4% and 8% oil was digested, respectively. Previous researchers have observed a similar phenomenon, where it was attributed to a decrease in the ratios of lipase-to-oil phase, calcium-to-free fatty acids, and/or calcium-to-bile salts with increasing lipid concentration [25,36].

### 2.4. Pesticides Bioaccessibility

The influence of pesticide hydrophobicity and oil droplet concentration on the bioaccessibility of pesticides on tomatoes was studied. Three pesticides with different hydrophobic characteristics were used in these studies: bendiocarb (log P = 1.7); parathion (log P = 3.8); chlorpyrifos (log P = 5.3). Both droplet concentration and pesticide hydrophobicity impacted the fraction of pesticide solubilized in the gastrointestinal fluids after digestion (Figure 6). For instance, the bioaccessibility of bendiocarb was relatively high (>90%) in the absence of emulsion, and was largely independent of oil droplet concentration (Figure 6a). This pesticide always has a high bioaccessibility because it is not strongly hydrophobic, and therefore can dissolve directly in the water phase. Indeed, it is reported that the water-solubility of bendiocarb is around 260 mg/L at 25 °C (chemspider.com), which is considerably higher than the total amount in the small intestine phase (around 100 mg/L). Conversely, the bioaccessibility of parathion and chlorpyrifos both increased appreciably with increasing oil droplet concentration (Figure 6b,c). For instance, the bioaccessibility of parathion was 17.8%, 66.8%, and 72.0% and chlorpyrifos was 3.9%, 47.7%, and 71.0% for 0%, 4%, and 8% oil, respectively. In general, the bioaccessibility of the pesticides therefore decreased as their hydrophobicity increased when ingested at the same fat content. This is because these molecules are more lipophilic than bendiocarb and therefore have lower water-solubilities: 11 and 1.1 mg/L for parathion and chlorpyrifos (at 25 °C), respectively (chemspider.com). As a result, these more hydrophobic pesticides have to be solubilized within the non-polar interiors of the mixed micelles in order to be bioaccessible. As the oil droplet concentration increases, there are more free fatty acids and monoacylglycerols available to participate in mixed micelle formation with the bile salts and phospholipids [37]. Consequently, a higher amount of the hydrophobic pesticides can be solubilized in the mixed micelles as the droplet concentration in a co-ingested emulsion is raised. The fact that the more hydrophobic pesticides had a lower bioaccessibility may be because not all of them could be transferred through the water and into the interiors of the mixed micelles.

## 3. Materials and Methods

### 3.1. Materials

Fresh organic tomatoes and corn oil (Mazola) were both obtained from a nearby supermarket. Bendiocarb, parathion, chlorpyrifos, mucin, pepsin, porcine lipase, and porcine bile extract were purchased from the Sigma-Aldrich Company (St. Louis, MO, USA). The Log P values of the pesticides were predicted using the XLogP3 model available from PubChem [38,39,40]. Whey protein isolate (WPI) was purchased from Agropur (Le Sueur, MN). A laboratory-scale water-purification system was utilized to prepare the double distilled water (Nanopure Infinity system, Barnstaeas International, Dubuque, IA, USA). All chemicals utilized were of analytical grade.

### 3.2. Pesticide-Enriched Model Food Preparation

Fresh organic tomatoes were washed under running water for 1 min and wiped dry with tissue paper to remove any residual pesticides. Organic tomatoes were used for this study because they typically have a lower level of pesticides than conventional ones. The washed organic tomatoes were cut into small cubes (around 10 × 10 mm) and then a tomato puree was prepared by blending them with a kitchen homogenizer for 1 min. Afterward, the tomato puree was combined with the pesticide standard solution to mimic a pesticide-enriched sample. A pesticide standard solution was prepared by dissolving weighed standard samples in acetonitrile to reach final concentrations of 10, 10, and 50 ppm for bendiocarb, parathion, and chlorpyrifos, respectively. These final concentrations were used because they were 100-fold higher than the reported maximum residue (MRL) level allowed, [41,42,43] which made them easier to detect using the HPLC method after the digestion process.

### 3.3. Emulsion Preparation

An aqueous phase was produced by dissolving 1% *w*/*w* WPI into phosphate buffer solution (5 mM, pH 7.0). This sample was then passed through a filter (0.45 μm) to eliminate insoluble particles that might block the homogenizer. Emulsion premixes were produced by combining 10% *w*/*w* of corn oil with 90% *w*/*w* of aqueous phase using a high-shear mixer for 2 min (M133/1281-0, Biospec Products, Inc., ESGC, Switzerland). The droplet size was then reduced further by passing the emulsion three times through the interaction chamber (F20Y) of a microfluidizer (M110Y, Microfluidics, Newton, MA) at 11,000 psi. The fine emulsions produced were then diluted to either 8% or 4% *w/w* oil content with buffer solution before using.

### 3.4. Gastrointestinal Tract (GIT) Model

Equal masses of pesticide-contaminated food and emulsion were mixed together to form the initial samples. For the samples without oil, phosphate buffer (5 mM, pH 7) was utilized instead of an emulsion. An in vitro GIT model, described in detail in our previous study, was employed to follow the gastrointestinal fate and digestion profile of our samples [33]. Basically, this involved passing each sample through simulated mouth, stomach, and small intestine stages, and then measuring their characteristics at the end of each stage.

### 3.5. Particle Characterization

The properties of the colloidal particles within each sample were measured using the analytical methods described previously [33]. The surface-weighted mean particle diameter (*d*_32_) and particle size distribution (PSD) were determined by laser diffraction (Mastersizer 2000, Malvern Instruments, Worcestershire, UK). Prior to analysis, all samples were diluted with phosphate buffer (5 mM, pH 7) to reach an optimum light scattering level, except the stomach samples, which were diluted with water acidified with HCl solution to reach pH 2.5. Refractive index values of 1.47 and 1.33 were used for the oil and water phases in the light scattering calculations [44].

The ζ-potential of the colloidal particles was determined by electrophoresis (Zetasizer Nano ZS series, Malvern Instruments). Again, colloidal dispersions were diluted with either phosphate buffer or acidified water prior to analysis to reach the optimum light scattering level of the instrument.

A digital camera attached to a confocal microscope was used to acquire images of sample microstructure at a magnification of 400× (Nikon, Melville, NY). Nile red dissolved in ethanol (1 mg/mL) was mixed into the samples (20:1 *w*/*w*) to stain the oil phase. The excitation wavelength for Nile red was 543 nm, whereas the emission wavelength was 605 nm. Images were collected, stored, and processed utilizing the instrument software (NIS-Elements, Nikon, Melville, NY).

### 3.6. Pesticide Bioaccessibility

Pesticide bioaccessibility was measured using an analytical protocol described in detail in a previous publication, with some adaptations [33]. Each sample was put through the entire simulated GIT and then the total digest was collected. A fraction of this digest was then centrifuged and the clear upper layer was taken to be the mixed micelle phase. Pesticide bioaccessibility was determined by measuring the pesticide concentrations in the mixed micelle phase (C_Micelle_) and in the total digest (C_Digest_):(1)$$Bioaccessibility\;\left(\%\right)=100\times \frac{C_{Micelle}}{C_{Digest}}.$$

### 3.7. Pesticide Quantification

The concentration of pesticides within the samples was determined using a solvent-extraction/chromatography method. The solvent-extraction step involved mixing 5 mL of total digest or mixed micelle phase with 2.5 mL of dichloromethane, vortexing for 1 min, then centrifuging at 4000 rpm for 5 min. The resulting supernatants were then collected and the samples were extracted one more time using the same method. The collected supernatants were then combined and dried under nitrogen in a dark environment. The residues were then dissolved in acetonitrile to an appropriate concentration and then filtered through a 0.45 µm PTFE filter.

The HPLC system used in this study consisted of a binary pump, an online degasser, an auto-sampler, a temperature controller, and a UV detector (Agilent, Santa Clara, CA). A reversed-phase column (Zorbax SB-C18, 250 mm × 4.6 mm id, 5 μm) was used to separate various pesticides employing a mobile phase that consisted of acetonitrile: 0.1% phosphoric acid (70:30) at the flow ratio of 1 mL/min. An injection volume of 20 µL and an operating temperature of 30 °C were used. The bendiocarb, parathion, and chlorpyrifos were detected at wavelengths of 210, 275, and 288 nm, respectively. The chromatographs obtained were analyzed by the instrument software (Agilent ChemStation, Santa Clara, CA) and the pesticide concentrations were then calculated using standard curves.

The three pesticides used in this study were well separated within 20 min using the HPLC chromatography method employed (Figure 7). The standard curve established between the detector voltage and pesticide concentration showed good linearity and a high correlation (R^2^ > 0.99) for all pesticides. The detection range was 0.5–25 ppm for bendiocarb and parathion, and 2.5–125 ppm for chlorpyrifos. The recovery was 105.4%, 82.9%, and 87.5% for bendiocarb, parathion, and chlorpyrifos, respectively.

### 3.8. Statistical Analysis

Three or more fresh samples were prepared for each experiment. Data are presented as means and standard deviations calculated using statistical software (SPSS). A Duncan’s test was employed to establish whether the difference between mean values was statistically significant (*p* < 0.05).

## 4. Conclusions

In summary, our in vitro results suggest that the bioaccessibility of a pesticide on fresh produce depends its hydrophobicity, as well as the amount of oil droplets co-ingested with it. The bioaccessibility of a polar pesticide (low Log P) is relatively high and does not depend strongly on oil droplet concentration because it is readily soluble in water. Conversely, the bioaccessibility of a non-polar pesticide (high Log P) is relatively low in the absence of oil droplets, but increases appreciably when the oil droplet concentration is raised because it can only be solubilized in the hydrophobic core of mixed micelles. In future studies, it would be advantageous to determine the impact of pesticide hydrophobicity and oil droplet concentration on the bioavailability and bioactivity of pesticides using in vivo studies, such as an animal feeding model. It may then be possible to establish in vivo–in vitro correlations that could guide the formulation of emulsions that could increase the bioavailability of beneficial bioactives but not detrimental ones.

## Figures and Tables

**Figure 1 molecules-25-01466-f001:**
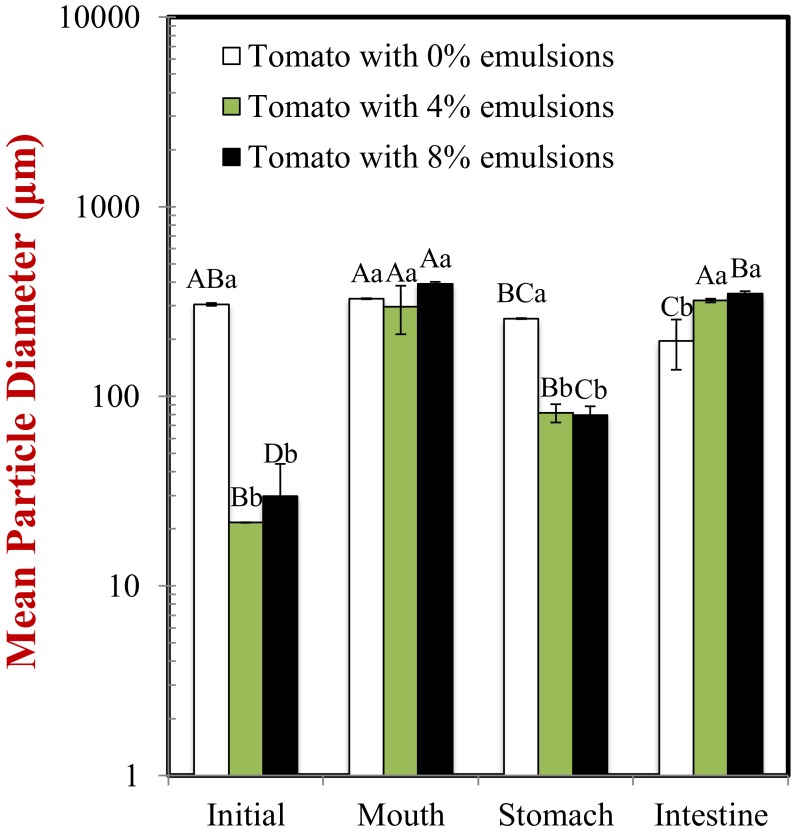
Changes in the surface-weighted mean particle diameter (*d*_32_) of samples as they passed through a simulated gastrointestinal tract. Samples designated with upper-letters (**A**, **B**, **C**) were significantly different (Duncan, *p* < 0.05) when compared between different GIT conditions (same lipid content). Samples with different lower-case letters (a, b, c) were significantly different (Duncan, *p* < 0.05) when compared between different lipid content (same GIT condition).

**Figure 2 molecules-25-01466-f002:**
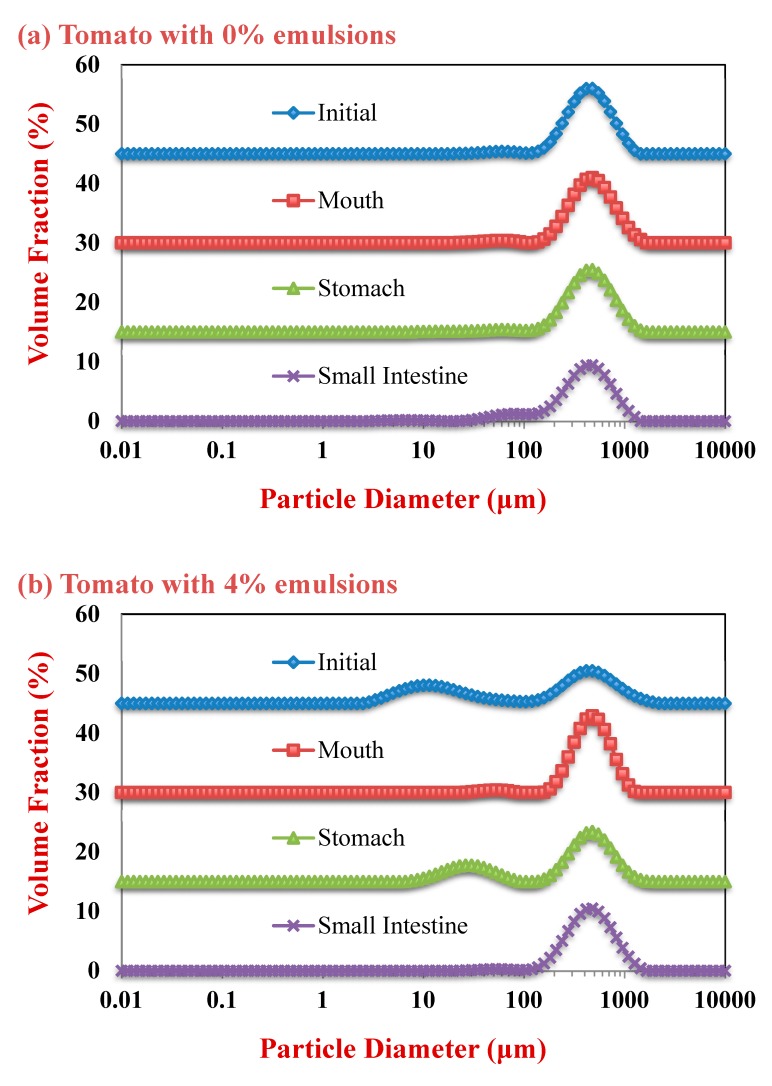
Particle size distributions of samples with different lipid contents as they were exposed to simulated gastrointestinal tract (GIT) conditions: (**a**) tomato with 0% emulsions; (**b**) tomato with 4% emulsions; and, (**c**) tomato with 8% emulsions.

**Figure 3 molecules-25-01466-f003:**
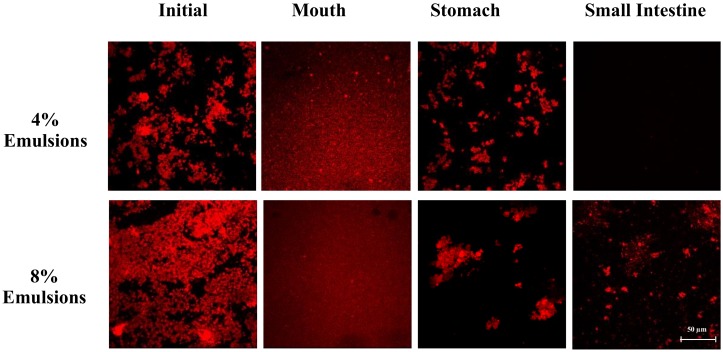
Microstructure of samples with different lipid contents after they were incubated under different GIT conditions.

**Figure 4 molecules-25-01466-f004:**
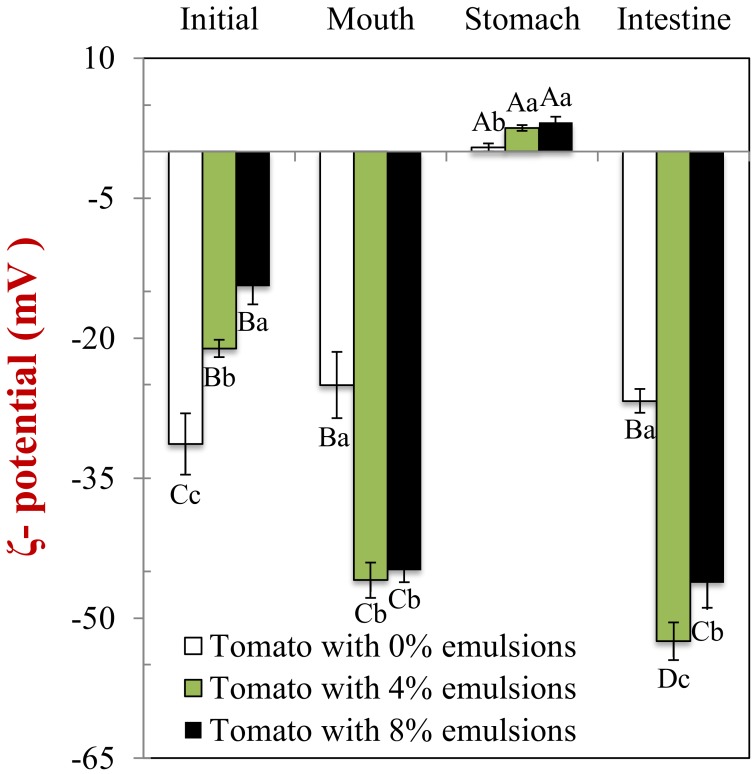
The electrical characteristics (ζ-potentials) of samples containing different lipid content. Samples designated with upper-letters (**A**, **B**, **C**) were significantly different (Duncan, *p* < 0.05) when compared between different GIT conditions (same lipid content). Samples with different lower-case letters (a, b, c) were significantly different (Duncan, *p* < 0.05) when compared between different lipid content (same GIT condition).

**Figure 5 molecules-25-01466-f005:**
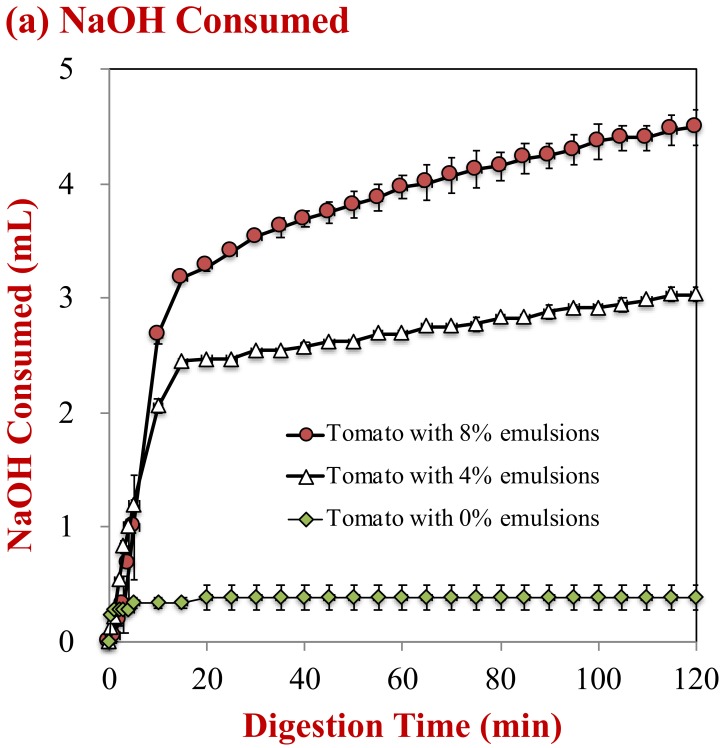
Digestion profiles of samples with different lipid contents during simulated digestion in the small intestine: (**a**) Amount of NaOH consumed; and, (**b**) the fatty acid release profile.

**Figure 6 molecules-25-01466-f006:**
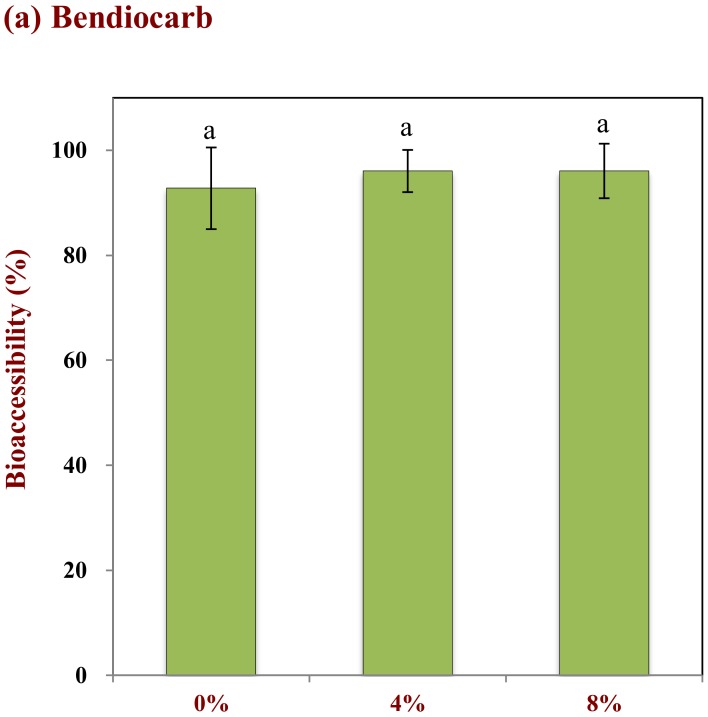
Impact of the lipid content on the bioaccessibility of pesticides with different Log P values: (**a**) Bendiocarb (log P = 1.7), (**b**) parathion (log P = 3.8), and (**c**) chlorpyrifos (Log P = 5.3). Samples designated with different capital letters (**a**, **b**, **c**) were significantly different (Duncan, *p* < 0.05) when compared between different lipid contents.

**Figure 7 molecules-25-01466-f007:**
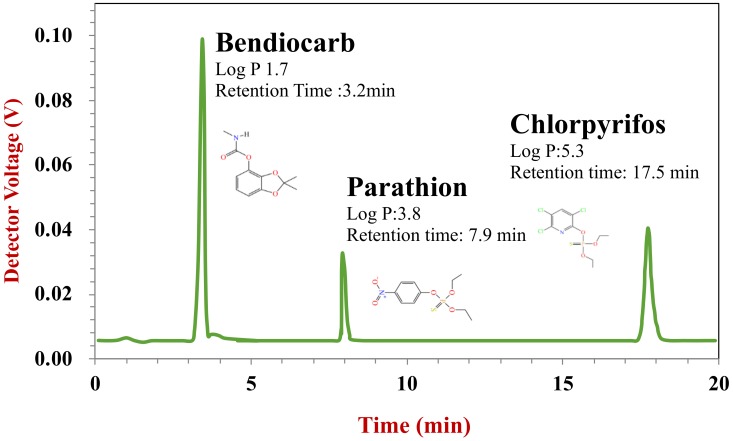
HPLC chromatogram of mixed pesticides standard. Key to peak identity: 1, bendiocarb; 2, parathion; and 3, chlorpyrifos.

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
