# Peer review of "Impact of Pesticide Type and Emulsion Fat Content on the Bioaccessibility of Pesticides in Natural Products"

_molecules, 2020, doi:10.3390/molecules25061466_

Round 1

Reviewer 1 Report

In this article, authors have examined the influence of pesticide type on the bioaccessibility of pesticides on tomatoes co-ingested with a model food emulsion. Three representative pesticides were measured after samples were passed through a simulated gastrointestinal tract to study the impact of the oil content of emulsion based foods on pesticide bioaccessibility. Results provide insights into the influence of properties of pesticides on their bioaccessibility in the presence of emulsion-based foods.

The paper is clear, understandable and appropriate for publication in Molecules. I have a minor comment: Authors should be more careful with the level of self-citation in the references section (13 from a total of 34 ). Please, modify and add new references with other authors.

Based on all above comments, I propose this manuscript to be reconsider after this minor revision.

Author Response

Responses to Reviewer 1

- We thank the reviewer for their constructive comments and suggestions on our manuscript.  We have revised it according to these comments and included a detailed list of responses below.

In this article, authors have examined the influence of pesticide type on the bioaccessibility of pesticides on tomatoes co-ingested with a model food emulsion. Three representative pesticides were measured after samples were passed through a simulated gastrointestinal tract to study the impact of the oil content of emulsion-based foods on pesticide bioaccessibility. Results provide insights into the influence of properties of pesticides on their bioaccessibility in the presence of emulsion-based foods.

The paper is clear, understandable and appropriate for publication in Molecules. I have a minor comment: Authors should be more careful with the level of self-citation in the references section (13 from a total of 34). Please, modify and add new references with other authors.

  • We thank the author for their positive comments on our manuscript. The reason for a high level of citation is that the authors have worked extensively in this area for over a decade and have published many (> 100) articles on food matrix effects and bioaccessibility.  Our previous work has mainly been on enhancing the bioaccessibility of beneficial nutrients, whereas this work shows that emulsions may also enhance the bioaccessibility of pesticides.  Nevertheless, we have revised the manuscript based on the reviewer’s comments to include a number of additional articles from other researchers in this area.

Based on all above comments, I propose this manuscript to be reconsider after this minor revision.

Reviewer 2 Report

Specific Comments:

The topic is worthy of study and appropriate methods, in general, are described. The study would have been stronger had multiple concentrations of the 3 pesticides been evaluated; and the justification for this selection of a relatively high residue level as the single concentration of 100x the reported maximum residue (MRL) level because this concentration was easier to detect using their HPLC method.

The authors must explain why, at 4% oil concentration, the bioaccessibility of parathion (log P = 3.8) is significantly greater than that of chlorpyrifos (log P = 5.3) in the Discussion.

The stated objective of “in vivo – in vitro correlations that could guide the formulation of emulsions that could increase the bioavailability of beneficial bioactives but not detrimental ones” seems plausible when 2 single compounds each with a defined set of physicochemical properties are considered, but less so when mixtures of lipophilic xenobiotics are considered as the physicochemical properties of the beneficial vs detrimental chemicals are likely to overlap.

General Comments:

Line 31: “--- undesirable non-polar ---” might be replaced with “--- undesirable lipophilic ---”

Line 32: “--- such as pesticides” might be replaced with “including pesticides”.

Line 38: --- emulsion-based products are --- (are requires a plural noun).

Line 40:  non-polar nutraceuticals might be lipophilic nutraceuticals? The latter is more definitive to this reviewer but either is correct.

Line 43: --- effects of emulsions is are influenced ---

Line 46:  If these lipid-containing emulsions enhance the bioavailability of lipophilic nutraceuticals in agricultural products how could they not also increase the bioavailability of lipophilic pesticides?

Line 63: ---pesticide type on the its bioaccessibility of pesticides ---; of pesticides is redundant.

Line 92: Is 1 wt% whey protein isolate 1%w/v or 1%w/w WPI? Scientifically, I suggest this concentration is better expressed as %w/v or %w/w here and throughout the manuscript.

Line 93: “--- passed through a filter to eliminate insoluble particles ---. What was the size of the filter?

Line 112: What acid(s) were used to achieve pH 2.5? This might be stated.

Line 125: Is this the entire GIT; or the entire simulated GIT? If the latter, this should be clearly sated.

Line 147: --- were adequately separated should be changed to were well separated

Line 160: Results and Discussion, not Discussions

Line 170: --- mixed with the tomatoes or mixed with the tomato puree?

Line 219: This suggest suggests that the mucin ---

Line 304: -- not strongly hydrophobic is awkward. Why not use hydrophilic?

Line 308: Rather than n=than “are strongly hydrophobic” use are more lipophilic than bendiocarb.  In short, log P is not a measure of not strongly hydrophobic.

Figure 7, line 319: How do the authors explain the fact that at 4% oil the bioaccessibility of parathion (log P = 3.8) is significantly greater than that of chlorpyrifos (log P = 5.3). Clearly, this is following log P. The explanation for this digression from their working hypothesis should be clearly explained in the Discussion.

Line 332: The stated objective of “in vivo – in vitro correlations that could guide the formulation of emulsions that could increase the bioavailability of beneficial bioactives but not detrimental ones” seems plausible when 2 single compounds each with a defined set of physicochemical properties are considered, but less so when mixtures of lipophilic xenobiotics are considered as the physicochemical properties of the beneficial vs detrimental chemicals are likely to overlap.

Author Response

Responses to Reviewer 2

- We thank the reviewer for their constructive comments and suggestions on our manuscript.  We have revised it according to these comments and included a detailed list of responses below.

The topic is worthy of study and appropriate methods, in general, are described. The study would have been stronger had multiple concentrations of the 3 pesticides been evaluated; and the justification for this selection of a relatively high residue level as the single concentration of 100x the reported maximum residue (MRL) level because this concentration was easier to detect using their HPLC method.

- The main focus of the current study was to examine the impact of pesticide type and oil concentration on the bioaccessibility of the pesticides.  In future studies, it would be interesting to examine the impact of pesticide concentration on bioaccessibility, but this was beyond the scope of the current study.  We thank the reviewer for this suggestion.

The authors must explain why, at 4% oil concentration, the bioaccessibility of parathion (log P = 3.8) is significantly greater than that of chlorpyrifos (log P = 5.3) in the Discussion.

 - As requested by the reviewer, we have revised the manuscript to discuss the differences in bioaccessibility for these two pesticides. (Please see “Pesticides Bioaccessibility” Section).

The stated objective of “in vivo – in vitro correlations that could guide the formulation of emulsions that could increase the bioavailability of beneficial bioactives but not detrimental ones” seems plausible when 2 single compounds each with a defined set of physicochemical properties are considered, but less so when mixtures of lipophilic xenobiotics are considered as the physicochemical properties of the beneficial vs detrimental chemicals are likely to overlap.

 - We agree with the reviewer, that the bioaccessibility of different kinds of molecules may be increased to different extents.  However, a better fundamental understanding of the origin of these effects could help formulate emulsion-based foods that could help promote beneficial ones, without promoting detrimental ones (but this is very system dependent).

General Comments:

Line 31: “--- undesirable non-polar ---” might be replaced with “--- undesirable lipophilic ---”

  • Revised

Line 32: “--- such as pesticides” might be replaced with “including pesticides”.

  • Revised

Line 38: --- emulsion-based products are --- (are requires a plural noun).

  • Revised

Line 40:  non-polar nutraceuticals might be lipophilic nutraceuticals? The latter is more definitive to this reviewer but either is correct.

  • Revised

Line 43: --- effects of emulsions is are influenced ---

  • Revised

Line 46:  If these lipid-containing emulsions enhance the bioavailability of lipophilic nutraceuticals in agricultural products how could they not also increase the bioavailability of lipophilic pesticides?

  • It depends on the molecular dimensions of the nutraceuticals and pesticides – they have to be big enough to fit into the hydrophobic domains of the mixed micelles.

Line 63: ---pesticide type on the its bioaccessibility of pesticides ---; of pesticides is redundant.

  • Revised

Line 92: Is 1 wt% whey protein isolate 1%w/v or 1%w/w WPI? Scientifically, I suggest this concentration is better expressed as %w/v or %w/w here and throughout the manuscript.

  • Revised to %w/w

Line 93: “--- passed through a filter to eliminate insoluble particles ---. What was the size of the filter?

  • The filter pore size (0.45 mm) has been included in the revised manuscript.

Line 112: What acid(s) were used to achieve pH 2.5? This might be stated.

  • As suggested this information had been included (HCl).

Line 125: Is this the entire GIT; or the entire simulated GIT? If the latter, this should be clearly sated.

  • As suggested the fact a simulated GIT was used has been stated in the revised manuscript.

Line 147: --- were adequately separated should be changed to were well separated

  • Revised

Line 160: Results and Discussion, not Discussions

  • Revised

Line 170: --- mixed with the tomatoes or mixed with the tomato puree?

  • Revised

Line 219: This suggest suggests that the mucin ---

  • Revised

Line 304: -- not strongly hydrophobic is awkward. Why not use hydrophilic?

  • The molecules are hydrophobic (logP > 1) but less than hydrophobic than the other pesticides. (They are not hydrophilic)

Line 308: Rather than n=than “are strongly hydrophobic” use are more lipophilic than bendiocarb.  In short, log P is not a measure of not strongly hydrophobic.

  • Revised as suggested.

Figure 7, line 319: How do the authors explain the fact that at 4% oil the bioaccessibility of parathion (log P = 3.8) is significantly greater than that of chlorpyrifos (log P = 5.3). Clearly, this is following log P. The explanation for this digression from their working hypothesis should be clearly explained in the Discussion.

  • As suggested, we have listed the bioaccessibility of the different pesticides in the revised manuscript and given a reason for their differences.

Line 332: The stated objective of “in vivo – in vitro correlations that could guide the formulation of emulsions that could increase the bioavailability of beneficial bioactives but not detrimental ones” seems plausible when 2 single compounds each with a defined set of physicochemical properties are considered, but less so when mixtures of lipophilic xenobiotics are considered as the physicochemical properties of the beneficial vs detrimental chemicals are likely to overlap.

See comment ab

Reviewer 3 Report

The manuscript is written in a correct english and only minor checks are required.

The experimental path is correct and the results are discussed.

When I download the pdf manuscript, figure 3 is not clear, with colored curves or trends completely shifted out. Authors should check

In figures 5 and 7 the error bars show an error that in my opinion is too high. 

The difficulty of the experiments can be understood but, in my opinion, the results should be more rigorous.

I suggest to add more results to support the publication.

Author Response

- We thank the reviewer for their constructive comments and suggestions on our manuscript.  We have revised it according to these comments and included a detailed list of responses below.

The manuscript is written in a correct english and only minor checks are required.

  • We thank the reviewer for their positive comments.

The experimental path is correct and the results are discussed.

  • Again, we thank the reviewer for their positive comments.

When I download the pdf manuscript, figure 3 is not clear, with colored curves or trends completely shifted out. Authors should check

  • We apologize for this problem – we have checked the figure and uploaded it in PDF format in the revised manuscript.

In figures 5 and 7 the error bars show an error that in my opinion is too high. The difficulty of the experiments can be understood but, in my opinion, the results should be more rigorous.

  • The error bars were calculated from multiple measurements made on freshly prepared samples, and are typical for this type of experiment where complex biological samples are being analyzed (tomatoes with pesticides on). When we use more well-defined samples (such as carotenoid-loaded nanoemulsions), then we get less experimental errors.  The error bars are just a consequence of the variability in the kind of sample analyzed.

I suggest to add more results to support the publication.

  • The purpose of this stud was to examine the impact of pesticide type and oil concentration on the bioaccessibility of the pesticides. Consequently, we used three different pesticides with different polarities, and three different fat concentrations to represent different foods.

Round 2

Reviewer 3 Report

Unfortunately I still see figures 3 with shifted curves

However, if it is only my problem, doesn't matter.

Suggestions have been considered.

My opinion is to accept the manuscript